# Clinical Utility of Ocular Assessments in Sport-Related Concussion: A Scoping Review

**DOI:** 10.3390/jfmk9030157

**Published:** 2024-09-04

**Authors:** Ayrton Walshe, Ed Daly, Lisa Ryan

**Affiliations:** Department of Sports, Exercise, and Nutrition, Atlantic Technological University, H91 T8NW Galway City, Ireland; ayrton.walshe@research.atu.ie (A.W.); ed.daly@atu.ie (E.D.)

**Keywords:** ocular, technology, mTBI, eye-tracking, vision, concussion, athletes, military

## Abstract

**Background/objectives:** Ocular tools and technologies may be used in the diagnosis of sport-related concussions (SRCs), but their clinical utility can vary. The following study aimed to review the literature pertaining to the reliability and diagnostic accuracy of such assessments. **Methods:** The preferred reporting items for systematic reviews and meta-analysis (PRISMA) extension for scoping reviews was adhered to. Reference standard reliability (RSR ≥ 0.75) and diagnostic accuracy (RSDA ≥ 0.80) were implemented to aid interpretation. **Results:** In total, 5223 articles were screened using the PCC acronym (Population, Concept, Context) with 74 included in the final analysis. Assessments included the King-Devick (KD) (n = 34), vestibular-ocular motor screening (VOMs) and/or near point of convergence (NPC) (n = 25), and various alternative tools and technologies (n = 20). The KD met RSR, but RSDA beyond amateur sport was limited. NPC met RSR but did not have RSDA to identify SRCs. The VOMs had conflicting RSR for total score and did not meet RSR in its individual tests. The VOMs total score did perform well in RSDA for SRCs. No alternative tool or technology met both RSR and RSDA. **Conclusion:** Ocular tools are useful, rapid screening tools but should remain within a multi-modal assessment for SRCs at this time.

## 1. Introduction

Sport-related concussions (SRCs) are a mild traumatic brain injury whereby forces are transmitted to the brain via direct head contact or transmission of forces via contact elsewhere in the body [1]. Such an injury induces a metabolic cascade and transient symptomology in athletes, the duration of which is variable but typically occurs within a month [1]. Although these symptoms are undesirable, they do provide useful insights for clinicians to initially diagnose and even provide tailored rehabilitation to patients [2]. Both quantitative and qualitative research have shown a lack of SRC reporting in some sports and difficult biopsychosocial barriers to best practice in SRC management [3,4].

The recommended SRC assessment of athletes at this time is the Concussion in Sport Group’s Sport Concussion Assessment Tool 6 (SCAT6) [5,6,7]. However, this multi-modal assessment can only be conducted by medical professionals and can be difficult to conduct in non-specialized environments [8]. Simplified sideline assessments and emerging technologies have attempted to bridge this gap between specialized and non-specialized (i.e., community athlete care) through the inception of neuropsychological [9], haematological [10], and ocular assessment tools [11].

Ocular tools are one of the most commonly used SRC assessments, with their development originally extending as far back as the 1970s [12,13]. These tools assess a variety of ocular movements such as saccades, smooth, pursuits, near point of convergence (NPC), vestibular-ocular reflexes, and visual motion sensitivity. Two of the most popular assessments in this area are the King-Devick (KD) test [14] and vestibular-ocular motor screening (VOMs) [15]. The VOMs is included within the office-based SCAT6, and both the KD and VOMs are included in National Collegiate Athletic Association—Department of Defence Concussion Assessment, Research and Education (NCAA DOD CARE) consort assessments, which collects and publishes a large body of work in SRC assessments in both student-athletes and military personnel [16]. The associated eye movements (e.g., saccades and smooth pursuits) of these tests are a key focus of emerging ocular technologies (primitive or untested) in SRC research [17,18]. Emerging technologies include new, untested technologies or technologies still within their infancy in research or practice.

Ensuring the reliability and validity of assessment tools is imperative to high-quality SRC diagnosis. Reliability is the repeatability or consistency of test measures [19] and is traditionally assessed through intra-class correlation coefficients (ICCs), but more traditional association tests may also be considered. Validity refers to how truthful a measure is [19] and is assessed using sensitivity (ability to identify a true positive, i.e., concussed) [20], specificity (ability to identify a true negative, i.e., non-concussed) [20], and receiver operator curves (ROC), which plot the relationship of both simultaneously [21]. These statistics allow researchers and practitioners to determine (a) the reproducibility of an assessment and (b) the diagnostic accuracy of an assessment. It is common for studies utilising traditional ocular assessments (VOMs and KD) to cite a limited number of seminal studies for reliability or diagnostic accuracy statistics [14,15,22], while little is known about the reliability and diagnostic accuracy of emerging ocular technologies.

With this in mind, using a scoping review methodology, the following study aimed to explore and summarize current literature on the reliability and diagnostic accuracy of such ocular tools used in the diagnosis of SRCs in athletes and military personnel. This study aims to guide further research in sport and emerging technology development and to quantify the presence and clinical utility of all available ocular tools. Thus, given the broad research aim, a scoping review was deemed appropriate.

## 2. Materials and Methods

Ethical approval was not required for this study. Registration was completed on Open Science Framework (OSF) on 1 November 2023 as advised by the Joanne Briggs Institute [23], and this study is reported in accordance with the modified preferred reporting information for systematic reviews and meta-analysis protocol for scoping reviews (PRISMA-SCr) [24]. Please see Appendix B for further reporting detail.

The key research question supporting this study was: Is there research to support the use of ocular assessment tools commonly used to diagnose SRCs in athletes? More specifically, this study had three questions to answer: Which ocular tools provide (1) optimal reliability, (2) internal consistency, and (3) diagnostic accuracy in the diagnosis of SRCs.

Studies were identified via four databases (SPORTDiscus, PubMed, Web of Science, and CINAHL). Initial search terms and strings were conducted by AW and were based on a previous systematic review on ocular technologies for SRCs, which investigated variables and measures of interest (but not reliability and diagnostic accuracy) for future research [25]. These were later refined based on the initial searches, and agreement was reached on the final search terms by AW, LR, and ED on 26 October 2023. The final search was conducted on 2 November 2023. The included search terms were as follows: Concussion or Brain injur* OR Head injur* OR Athletic injur* OR “Sports concussion” OR “Sports related concussion” OR “mild traumatic brain injury” OR mTBI OR TBI OR “craniocerebral trauma” OR “brain damage” OR SRC AND Athlete* OR Sport* OR Player* OR “Physically Active” OR Healthy OR Active OR Rugby OR Soccer OR Football OR Gaelic Football OR Camogie OR Hurling OR Hockey OR AFL OR NFL OR “Australian Rules” AND Eye OR “Eye movement” OR Eye track* OR Gaze track* OR Oculomotor OR “Pupil dilation” OR “Pupil size” OR Pupillometer OR Pupillometry OR Saccad* OR “Smooth pursuit” OR Visuomotor OR “saccadic dysfunction” OR “saccadic eye movement” OR vestibular OR ocular OR “ocular microtremor” OR “rapid eye movements” OR “Near point of convergence” OR “Balance vision reflex” OR “Visual motion sensitivity” OR “Contrast sensitivity”.

Appropriate studies were selected using the Population, Concept, Context (PCC) protocol for inclusion and exclusion criteria. Inclusion criteria were as follows: otherwise healthy athletes or active military personnel (with or without an SRC) involved in observational, cross-sectional, case-control, or intervention-based research whereby SRC-related ocular assessments were conducted; reliability, sensitivity/specificity, or ROC data were reported; published between 2000–2023; and published in English. Studies were to be excluded if the participants were not specifically defined as athletes or active military personnel or if their mTBI was not confirmed to be sport-related, if the study design was computer-modelled, qualitative, involved ocular assessment not specifically related to SRC or associated head impacts, did not report reliability, internal consistency, sensitivity and/or specificity data; and if the research was non-English or published outside the year range stated above. Both full text articles and conference abstracts were included to adequately represent emerging technology research, increase the comprehensiveness of our findings, and reduce reporting bias [26].

Study searches were exported to Endnote Desktop V.20 (Clarivate, London, UK) and screened in three phases for inclusion and exclusion; phase one (title screening), phase two (abstract screening), and phase three (full text screening). A coding hierarchy was applied using the ‘Research Notes’ function on Endnote to exclude studies based on duplicates (SRD), language (SRL), year (SRY), population (SRP), outcome (SRO), not a study of interest (SRNS), or miscellaneous (SRM). An additional code (SRR) was retrospectively added, as retracted articles were identified in the output. To confirm agreement on the application of the inclusion criteria and subsequent coding, a random sample of 25 articles were independently coded by AW and LR before consensus was reached. AW conducted all other screening but used the SRM code to discuss issues that arose during each phase. An overview of the screening process is available in Figure 1. Grey literature and citation searches were conducted to ensure all relevant studies were included in the analysis, and corresponding authors were conducted via email or via ResearchGate to obtain articles where required.

Data extraction was conducted using a modified version of a data extraction tool previously used by this research team [3] in Excel 2023 (Microsoft, Redmond, WA, USA). Specifically, amendments were made to obtain greater detail on each ocular tool and statistical analysis. Data were extracted regarding study design, setting, population, demographics, ocular tools, statistics reported, and any additional data deemed relevant. Data extraction was conducted by AW with oversight by LR and ED, and issues were discussed and resolutions agreed upon weekly throughout this process. Data were collated into a single Microsoft Excel spreadsheet for summary and subgroup analysis. Risk of bias for internal consistency and reliability was assessed using the Consensus-Based Standards for the Selection of Health Measurement Instruments (COSMIN) tools [27]; the internal consistency tool has five key questions while the reliability tool has eight. The revised tool for the Quality Assessment of Diagnostic Accuracy Studies (QUADAS-2) was used to assess diagnostic accuracy across four domains (patient selection, index tests, reference standards, flow and timing) related to the risk of bias and applicability concerns of studies [28]. No reference or gold standard of diagnosis currently exists for SRCs, thus medical diagnosis was used as the reference standard for this study.

Charted data was supported and interpreted using a number of classification criteria for related statistical analysis. Intraclass-correlation coefficients (ICC) are commonly used to assess the reliability of continuous variables. The ICC employed depends on model, type, and definition, and are graded as follows; ≤0.50 = poor, 0.50–0.75 = moderate, 0.75–0.90 = good, and >0.90 = excellent reliability [29]. Pearson’s (r) and Spearman’s (ρ) correlations are also commonly used for continuous data but generally not preferred in place of ICC. These statistics are interpreted as follows: 0.00–0.30 = negligible, 0.30–0.50 = low, 0.50–0.70 = moderate, 0.70–0.90 = high, and 0.90–1.00 = very high correlation [30]. Kappa (κ) is often used to assess agreement in qualitative or categorical data and thus is used in the VOMS assessment reliability analysis given the variability in symptom provocation in the test. It is interpreted as follows: <0 = no agreement; 0.01–0.20 = slight agreement; 0.21–0.40 = fair agreement; 0.41–0.60 = moderate agreement; 0.61–0.80 = substantial agreement; and 0.81–0.99 = almost perfect agreement. The reference standard of reliability (RSR) for clinical utility was set at 0.75 [31,32]. Internal consistency allows for the contribution of individual subscales of an instrument assess the various aspects of a construct. It is represented via Cronbachs alpha (*α*) and is interpreted as follows: <0.50 = unacceptable, 0.50–0.60 = poor, 0.60–0.70 = questionable, 0.70–0.80 = acceptable, 0.80–0.90 = good, and 0.90–1.00 = excellent [33]. As previously mentioned, sensitivity is the ability to identify a true positive in diagnosis, while specificity is the ability to identify a true negative in diagnosis. Sensitivity and specificity values are interpreted as follows: 0.80–0.89 = acceptable and 0.90–1.00 = good to excellent [34]. As a general rule, ROC area under the curve (AUC) analysis allows for the assessment of classifier performance and an exploration of the trade-off between sensitivity and specificity. AUC scores are interpreted as follows: <0.50 were = poor; 0.51 to 0.69 = fair; 0.70 to 0.80 = acceptable; 0.80 to 0.90 = excellent; and ≥0.90 = outstanding [35,36]. The reference standard of diagnostic accuracy (RSDA) for clinical utility was set at 0.80 [37,38]. Significance was set at *p* < 0.05 for all analyses.

## 3. Results

### 3.1. Study Characteristics

In total, 5223 articles were screened, with 71 full articles and three abstracts [39,40,41] included in the final analysis. The majority of studies were conducted in the USA (n = 59), with a large emphasis on high school and collegiate populations (n = 32). Included studies used the KD (n = 34), the VOMs in conjunction with independent use of the NPC assessment (n = 28), or alternative tools and emerging technologies (n = 19). Only four studies included active military personnel. Studies were mostly mixed sex (n = 51), and only one study included a female-only population. Findings for the KD, VOMS/NPC, and alternative tools and technologies are discussed in three succinct sections below. Further information on each study’s aims are available in Appendix A.

Risk of bias analysis for all assessed metrics is available in Appendix A. For risk of bias in internal consistency studies (n = 14), 10 were graded very good, three were graded doubtful, while one was graded unclear due to a lack of clarity as to whether Cronbach’s a was calculated for individual subscales (Appendix A). For risk of bias in reliability studies (n = 46), four were rated as very good, thirty-seven were graded as adequate, a further four were rated as doubtful, and one was inadequate (Appendix A). In analysis of risk of bias in diagnostic accuracy studies (n = 28), all but two studies met all applicability criteria (Appendix A). Studies were primarily exposed to potential bias due to convenience sampling, non-blinding of the index/reference standard findings (or not stating if such findings were known), or inappropriate follow up timelines following an SRC (e.g., retesting 21 days post-injury).

### 3.2. Clinical Utility of the King-Devick (KD) Assessment

Thirty-four studies conducted the KD test and reported on reliability (n = 23), internal consistency (n = 5), sensitivity and specificity (n = 10), and ROC analysis (n = 6). A summary of included studies is available in Table 1, and further information is available in Appendix A. Reliability of the KD via ICC was primarily good to excellent, with 94.12% (n = 48) of all included ICCs reaching the RSR (ICC = 0.75) for clinical utility. In instances where this threshold was not met, the retest comparison was of longer duration: pre-post season [42], median retest of 392 days [32], and between consecutive years [31]. The KD performed well in both card and digital versions, in the presence of exercise [43,44,45], combat sport [46,47], SRCs [43], and when administered by parents [47]. The KD also performed well with alternative reliability measures such as Pearson’s [48,49], Spearman’s [50,51], and Kappa [43,52] analyses, while internal consistency of the KD cards 1–3 was acceptable to excellent [14,53,54,55], even in the presence of oculomotor fatigue [41].

Findings in the assessment of diagnostic accuracy were not as clear. Using the ROC reference standard for clinical utility (AUC = 0.80), two studies observed outstanding diagnostic accuracy for the KD test, with a cutoff of 2 s time increase suggested to be optimal [56,57]. Meanwhile, another study found both the usage of the digital KD test (not card) and those with learning difficulty (but not those without) to have good diagnostic accuracy [36]. In comparison, the tool performed poorly in male elite rugby union athletes [58], and the utilization of eye tracking integration (KD-ET) would not be advised at this time (*p* > 0.05) [59,60]. Despite not meeting the RSDA, findings from Le et al. (2023) appear to show that the KD test may achieve higher diagnostic accuracy at 0–6 h/24–28 h post-SRC and decrease thereafter [36].

When sensitivity and specificity was assessed in isolation, the KD performed well in amateur rugby league and union [50,52] and in male sub-elite Australian Rules [43] but performed poorly in male elite rugby union [58] and semi-professional rugby union [61], and had low sensitivity in elite American football [45]. When exploring optimal cutoff times, any increase in KD time seemed to have the greatest combined sensitivity and specificity [32], with decreasing sensitivity but increasing specificity when >3 s or >5 s cutoffs were used. Comparatively, Le et al. explored optimal cutoffs to achieve sensitivities of 0.70 and 0.80, and this resulted in poor specificity at all time points, and no optimal cutoff existed once athletes became asymptomatic [36].

**Table 1 jfmk-09-00157-t001:** Overview of included studies using the King-Devick (KD) assessment.

Citation	Population	Sample Size (% Female)
Galetta et al., 2011a [46]	Amateur Boxing/MMA	39 (2.56%)
Galetta et al., 2011b [14]	Collegiate Multi-Sport	219 (16.89%)
King et al., 2012 [53]	Amateur Rugby League	50
King et al., 2013 [54]	Amateur Rugby Union	37
Leong et al., 2013 [47]	Amateur Boxing	33 (12.12%)
Yevseyenkov et al., 2013 [39]	High School American Football	47
Galetta et al., 2015 [56]	Youth/Collegiate Multi-Sport	322 (18.67%)
King et al., 2015a [50]	Amateur Junior Rugby League	19 (23.62%)
King et al., 2015b [52]	Amateur Rugby Union/League	104
Leong et al., 2015 [62]	Collegiate Football/Basketball	127 (6.30%)
Vartiainen et al., 2015 [48]	Professional Ice Hockey	185
Alsalaheen et al., 2016 [63]	High School American Football	62
Smolyansky et al., 2016 [64]	Elite Junior Olympians	54 (43.00%)
Walsh et al., 2016 [51]	Active Military	100 (21.00%)
Dhawan et al., 2017 [57]	High School Ice Hockey	141
Oberlander et al., 2017 [65]	Adolescent Athletes	68 (39.71%)
Weise et al., 2017 [66]	Adolescent Multi-Sport	619 (25.3%)
Broglio et al., 2018 [31]	Collegiate Multi-Sport/Active Military	4874 (41.09%)
Hecimovich et al., 2018b [43]	Sub-Elite Australian Football	22
Moran and Covassin 2018a [55]	Youth American Football/Soccer	422 (34.12%)
Worts et al., 2018 [44]	High-School Multi-Sport	45 (46.67%)
Breedlove et al., 2019 [67]	Collegiate Multi-Sport	3248 (44.70%)
Fuller et al., 2019 [58]	Elite Rugby Union	261
Hecimovich et al., 2018a [59]	Youth Australian Football	19
Naidu et al., 2018 [45]	Elite American Football	231
King et al., 2020 [68]	Amateur Rugby Union	69 (100%)
Molloy et al., 2017 [61]	Semi-Pro Rugby Union	52
Guzowski et al., 2017 [40]	American Football	124
White-Schwoch et al., 2019 [49]	Youth Tackle Football	82
Elbin et al., 2019 [42]	High School Multi-Sport	69 (46.40%)
Worts et al., 2020 [41]	Adolescent Multi-Sport	121 (41.32%)
Harmon et al., 2021 [32]	Collegiate Multi-Sport	82 (41.00%)
Hecimovich et al., 2022 [60]	Collegiate Rugby Union	49 (48.98%)
Le et al., 2023 [36]	Collegiate Multi-Sport	1559 (48.17%)

### 3.3. Clinical Utility of Vestibular-Ocular Motor Screening (VOMs) and/near Point of Convergence (NPC) Assessments

Twenty-eight studies included the NPC and VOMs assessment tools and reported on reliability (n = 16), internal consistency (n = 9), sensitivity and specificity (n = 4), and ROCs (n = 8). A summary of these studies is available in Table 2, and further information is available in Appendix A.

The NPC assessment had excellent intra-rater and inter-rater reliability via ICC [69,70,71] and Pearson’s correlations [72,73]. The within-session reliability of NPC also met the raw value threshold for clinical utility (ICC = 0.75), even in convergence insufficiency [74]; in various NPC font sizes [75]; and when compared across baseline, pre-practice, and intra-practice assessments [44]. However, the reliability of NPC reduced when observed over longer durations. Kontos et al. found moderate to good reliability between initial and 6-month follow ups for NPC distance and symptoms [22]. Despite this, the former reached clinical utility threshold and the latter was close behind. Additionally, poor to good one-year reliability for NPC and NPCbreak [76], poor to moderate NPC reliability over 18 months [77], and moderate reliability between consecutive years were observed [38]. Kappa analysis found fair agreement over 18 months for convergence [77], while fair and moderate agreement was observed for convergence and NPC distance, respectively, between consecutive years [31].

The VOMs total score had conflicting reliability via ICC, with one study achieving moderate to good reliability over 6 months [22] and achieving reference clinical utility, while another found poor reliability in consecutive years [38]. However, Kontos et al. (2020) also investigated subscales of the VOMs and found moderate to good reliability of visual motion sensitivity, vertical VOR, and vertical saccades to achieve RSR [22]. Agreement via Kappa was low, as values for individual subscales ranged from no agreement to fair [31,77].

Internal consistency of the VOMs total and NPC was found to be excellent in almost all instances [15,22,77,78,79,80], including in the presence of ocular motor fatigue [41]. Although Iverson et al. (2019) found that, in subscale investigations, only visual motion sensitivity and smooth pursuits had good consistency, while NPC and vertical saccades had acceptable consistency [80].

Due to the number of assessments within the VOMs, the investigation of diagnostic accuracy was more nuanced. Four studies found total VOM scores to be sufficient to achieve RSDA [38,81,82,83], while pretest symptoms also seemed to be accurate in the detection of SRCs [83]. The RSDA was not met in identifying normal versus protracted recovery in males or females [84]. VOMs change score achieved RSDA in one study [83] but performed worse in another when a cutoff of ≥3 was implemented [85]. A clinical cutoff of ≥4 was suggested to achieve RSDA for the VOMs and modified VOMs totals [83]; however, a higher AUC was achieved by Kontos et al. using a cutoff of ≥8 for VOMs total [82]. When looking at individual subscales as a diagnostic tool, one study found a combination of horizontal VOR, visual motion sensitivity, and NPC distance to be effective but did not find any individual subscale to reach the RSDA [15]. Poor subscale performance was also identified in mixed sex adolescent athletes [85]; however, both Kontos et al. (2021) and Ferris et al. (2022) found all subscales except NPC distance to have high diagnostic accuracy [82,83]. In fact, only two studies found convergence or NPC distance to have the RSDA for SRCs, suggesting limited utility of the tool to distinguish SRCs from controls [15,38,82,83,85,86].

**Table 2 jfmk-09-00157-t002:** Summary of included studies using the vestibular-ocular motor screening (VOMs) and/or near point of convergence.

**Vestibular-Ocular Motor Screening (VOMs)**		
**Citation**	**Population**	**Sample Size (% Female)**
Mucha et al., 2014 [15]	Unspecified Athletes	142 (34.51%)
Kontos et al., 2016 [78]	Collegiate Multi-Sport	263 (36.88%)
Broglio et al., 2018 [31]	Collegiate Multi-Sport/Active Military	4874 (41.09%)
Moran and Covassin 2018b [79]	Youth Multi-Sport	423 (34.28%)
Worts et al., 2018 [44]	High-School Multi-Sport	45 (46.67%)
Ferris et al., 2021a [81]	Collegiate Multi-Sport	388 (36.9%)
Iverson et al., 2019 [80]	Youth Ice Hockey	387
Kontos et al., 2020 [22]	Active Military	108 (14.8%)
Buttner et al., 2020 [87]	Adult Amateur	100 (28.00%)
Knell et al., 2021 [84]	Multi-Sport Athletes	549 (43.20%)
Kontos et al., 2021 [82]	Collegiate Multi-Sport/Cadet	570 (23.16%)
Elbin et al., 2022 [85]	Adolescent Multi-Sport	294 (42.52%)
Ferris et al., 2022 [83]	Collegiate Multi-Sport	3444 (47.80%)
Ferris et al., 2021b [38]	Collegiate Multi-Sport	3958 (47.70%)
Moran et al., 2023 [77]	Youth Soccer	51 (54.90%)
Anderson et al., 2024 [88]	Youth Soccer	110 (40.00%)
**Near Point of Convergence (NPC)**		
**Citation**	**Population**	**Sample Size (% Female)**
Pearce et al., 2015 [74]	Collegiate Athletes	78 (42.30%)
Kawata et al., 2015 [73]	Soccer Athletes	20 (25.00%)
Kawata et al., 2016 [72]	Collegiate American Football	29
McDevitt et al., 2016 [86]	Collegiate Multi-Sport	72 (41.67%)
DuPrey et al., 2017 [89]	Unspecified Athletes	270 (45.56%)
Aloosh et al., 2020 [76]	Elite Athletes	16 (56.25%)
Zonner et al., 2018 [69]	High School American Football	12
Worts et al., 2020 [41]	Adolescent Multi-Sport	121 (41.32%)
Heick et al., 2021 [75]	Recreational Multi-Sport	75 (78.67%)
De Rossi, 2022 [90]	High School Multi-Sport	718 (19.64%)
Kalbfell et al., 2023 [70]	Adult Soccer	43 (37.21%)
Zuidema et al., 2023 [71]	High School American Football	99

### 3.4. Clinical Utility of Alternative Tools and Technologies

Twenty studies reported reliability (n = 14), internal consistency (n = 1), sensitivity/specificity (n = 4), and ROCs (n = 8) on alternative assessments and emerging technologies. The primary eye movements assessed were saccades and smooth pursuits (n = 5) [32,76,91,92,93], dynamic visual acuity (DVA) (n = 4) [94,95,96,97], and ImPACT visual motor speed (VMS) (n = 7) [31,38,98,99,100,101,102]. A summary of included studies is available in Table 3, and further information is available in Appendix A.

ImPACT visual motor speed reliability was primarily assessed between consecutive years and either achieved the RSR [38,100] or had moderate to good reliability [31,101]. When observed over shorter durations, reliability was moderate to good between 7, 14, 30, 44, 198 days and only achieving the RSR at 7 and 44 days [99]. Impact VMS did not achieve the RSDA to distinguish SRCs versus controls [38,99,102], nor could it predict normal versus protracted recovery [102].

Five studies used emerging technologies to assess saccades and smooth pursuits (not including the KD eye-tracking previously stated). A proprietary algorithm for saccades could not reach RSR [76], while the neuro-otologic test chair only achieved the RSR when assessing optokinetic gain at 60° counter-clockwise [91]. In another study, optokinetic stimulation signs and symptoms achieved the RSDA, and this was further increased when used in conjunction with NPC [86]. A broad evaluation of SMI Red250Mobile found only the smooth pursuit saccade count at 10° to reach RSR in adults, while smooth pursuit diagonal gain, antisaccades, and fast memory guided sequences reached the RSR in youths [93]. However, the tool also had poor to questionable internal consistency. Two studies evaluated EYE-SYNC smooth pursuits. One study found the tool to not have the RSR or RSDA in tangential or radial variability [32]. In comparison, another study found both tangential and radial variability along with three other variables of EYE-SYNC to meet the RSR at both pre- and post-practice [92].

DVA reached the RSR in only one of three studies [94,95,96,97], and did not reach the RSDA for identifying SRCs versus controls [97]. PLM did not reach the RSDA across all variables explored [103], while no variable explored using the EyeLink 1000 reached the RSR for clinical utility [104]. Visio-vestibular evaluation (VVE) did not reach the RSDA across different rep ranges and movements; however, ≤20 repetitions was promising [105]. Gaze stability achieved the RSR in high-school students; however, it was only in the yaw plane and not in university students [95] and, although close, did not reach the RSDA using signs and symptoms [86]. No alternative tool or emerging technology met both RSR and RSDA.

**Table 3 jfmk-09-00157-t003:** Summary of included studies using alternative ocular assessments and emerging technologies.

Citation	Population	Sample Size (% Female)
**Dynamic Visual Acuity**		
Scherer et al., 2013 [94]	Active Military	20 (10.00%)
Kaufman et al., 2013 [95]	High School/Collegiate American Football	50
Patterson et al., 2017 [96]	Collegiate/Club Multi-Sport	28 (28.57%)
Feller et al., 2021 [97]	Collegiate Multi-Sport	86 (40.70%)
**Saccade and Smooth Pursuit Technologies**		
Cochrane et al., 2019 [91]	Collegiate Multi-Sport	115 (43.48%)
Sundaram et al., 2019 [92]	Collegiate Multi-Sport	150 (55.00%)
Aloosh et al., 2020 [76]	Elite	16 (56.25%)
Harmon et al., 2021 [32]	Collegiate Multi-Sport	82 (41.00%)
Sneigreva et al., 2021 [93]	Multi-Sport	92
**ImPACT Visual Motor Speed**		
Gardener et al., 2012 [98]	Non-Elite Rugby Union	92
Tsushima et al., 2016 [101]	Mixed Sex, High School	212 (40.57%)
Nelson et al., 2016 [99]	High School/Collegiate Multi-Sport	331 (16.62%)
Brett et al., 2016 [100]	High-School Multi-Sport	1150 (45.39%)
Sufrinko et al., 2017 [102]	Multi-Sport	69 (26.00%)
Broglio et al., 2018 [31]	Collegiate Multi-Sport/Active Military	4874 (41.09%)
Ferris et al., 2021b [38]	Collegiate Multi-Sport	3958 (47.70%)
**Additional Assessments**		
Kaufman et al., 2014 [95]	High School/Collegiate American Football	50
McDevitt et al., 2016 [86]	Collegiate Multi-Sport	72 (41.67%)
Howell et al., 2018 [104]	Adolescent Multi-Sport	31 (39.00%)
Master et al., 2020 [103]	Adolescent Multi-Sport	232 (57.33%)
Storey et al., 2022 [105]	Multi-Sport	138 (51.45%)

## 4. Discussion

This study aimed to summarize the available literature pertaining to the reliability and diagnostic accuracy of ocular assessment tools and emerging technologies used for SRC diagnosis in athletes and active military personnel. The findings are discussed further below in context of current literature.

### 4.1. Summary of Key Findings

The findings of this study confirm that ocular tools have potential in SRC diagnosis but may not be advised as stand-alone assessments at this time. The current literature in this area has a risk of bias due to convenience sampling, lack of blinding between reference and index standards, and the inherent subjectiveness of assessments such as the VOMs.

The KD had good to excellent reliability, achieving the RSR in almost all instances, while internal consistency was acceptable to excellent. Sensitivity and specificity was high in multiple studies but not in male semi-pro rugby union [61] or elite male rugby union [58]; the tool also had low sensitivity in male elite American Football. This may perhaps be linked to the rudimentary scoring system employed with the tool (total time), enabling inflated baselines from athletes in elite male athletes where poor attitudes towards this injury are prevalent [45,106]. Outstanding diagnostic accuracy and RSDA of the KD were observed using change from baseline [56] or an optimal cutoff of two seconds [57]. Excellent diagnostic accuracy and RSDA were observed using the digital KD and in those with learning disorders [36], while exploration of cutoff scores suggested using any increase in time, as >3 s and >5 s cutoffs had increasing specificity and decreasing sensitivity [32]. Le et al. (2023) [36] also explored cutoffs to achieve sensitivity at 70–80% across various time points post-SRC; however, this could not be achieved without obtaining low specificity [36]. Integrating eye tracking technology with the KD would not be advised at this time due to an inability to achieve high sensitivity, specificity, and the RSDA [59,60]. Ultimately, the findings indicate the KD may be a useful assessment to support suspected diagnosis in amateur sport settings.

NPC achieved the RSR (via ICC and Pearson’s r) in intra-rater [69,71,72,73], inter-rater [70,71], within-session [74,75], and post-exercise [44], but repeated testing may be required as reliability seemed to be reduced over longer durations [38,76,77]. VOMs total had poor reliability between years in mixed sex, collegiate athletes [38] but achieved the RSR with moderate to good reliability in military personnel [22]. Reliability via Kappa was mostly fair for VOMs subscales [31,77], while internal consistency of total VOMs and NPC was primarily excellent [15,22,41,77,78,79]; however, this did lower when individual subscales were assessed [80]. Although NPC had high reliability, limited ability to achieve RSDA to distinguish SRCs from controls were found [15,38,82,83,85], and NPC could not predict prolonged recovery [89]. VOMs total achieved the RSDA with excellent to outstanding diagnostic accuracy [38,81,82,83], but conflicting findings existed for VOMs subscales, as two studies achieved the RSDA [82,83] while another two did not [15,85]. However, it is difficult to truly compare these studies given the differences in the cutoffs implemented. Therefore, the NPC assessment should not be used to diagnose SRCs, but the VOMs assessment may be useful if reliability is ensured.

Additional information regarding the remaining alternative tools and technologies (as per Table 3) is available in Appendix A. However, many of these studies were exploratory or validation studies, and none had both of the required RSR and RSDA. Thus, none of these tools or technologies are advised for diagnosing SRCs at this time.

### 4.2. Findings in Context

SRCs are a challenging injury to accurately diagnose in amateur sport. This is exemplified by the majority of included studies being conducted in the US where legal incentives and structured programmes such as the CARE consortium [16] produce the majority of research available. Only three studies were conducted in Europe [87], and two of these were conducted on professional athletes [48,58].

The KD and VOMs/NPC are commonly implemented in pre-season baseline testing, particularly in US high school and collegiate athletes. Both tools are quick and easy to administer, but the VOMs does require clinical expertise to conduct. In comparison, the KD is cheap, requires minimal equipment, and was shown to have high reliability even when administered by sport parents [47]. If implemented, changes from baseline [56] or a cutoff of 2 s may provide the greatest accuracy for SRCs [57]. The tool may also not be effective in elite or semi-pro male athletes given the rudimentary scoring system, as previously mentioned [45,53,58]. This may be where an emerging technology such as the KD-ET could improve the diagnostic accuracy of the tool following further validation [59,60]. The tool might also only be effective for initial diagnosis, as diagnostic accuracy be reduced after 0–6 h/24–48 h post-SRC [36]. Similar findings have been observed in the SCAT assessment, whereby clinical utility can diminish after 72 h [1]. If medical expertise is available, the use of VOMs total seems to achieve high internal consistency and diagnostic accuracy. However, given the reliability concerns presented in this study, establishing intra- and inter-rater reliability is advised, as is the use of regular baseline testing to observe changes over time.

Emerging ocular technologies provide an attractive avenue for future research. Such tools could provide an objective assessment of SRCs from grassroots to professional sport. Achieving truly objective assessment is a pertinent point, as athletes may attempt to falsify assessment [107], while allied medical staff may feel pressure and bias from coaches or athletes during SRC assessments [4]. Research in athletic therapy students has also shown varying levels of self-efficacy in the assessment and management of concussion [108]. For instance, mean self-efficacy in the KD was 15.05 ± 30.30% with a 7.5 ± 20.34% use on clinical placement (r = 0.71, *p* < 0.001), while the VOMs obtained a mean self-efficacy of 57.71 ± 36.98% and a use of 31.44 ± 35.02% (r = 0.60, *p* < 0.001) [108]. Technologies may therefore help reduce the reliance and burden on medical professionals in the initial diagnosis, reduce the financial burden on athletes, parents, and sport clubs, and also reduce the burden on healthcare systems. The Noeye tracking technology included in the present study had achieved both sufficient reliability and validity for use to date, a finding consistent with a previous systematic review of eye tracking technology in SRCs [25]. The review did. However, suggest that challenging measures of executive function (i.e., memory-based saccades, antisaccades) may be a promising area for future research. Subsequent research by the same author has been included in the present study and is available in Appendix A. The study found that antisaccades using an emerging technology achieved moderate to good reliability in youth and adult participants, but many other explored variables performed poorly [93].

Why eye-tracking technologies have not been successful to date is unclear. It should be stated, firstly, that all oculomotor tools are indirect measures of brain function and thus are not as accurate as a more direct measure of brain injury. It could also be suggested that not all athletes who sustain an SRC present with oculomotor deficits, thereby impacting the diagnostic accuracy of such devices. As evidenced by the findings of these studies in Appendix A, it remains unclear which variables provide the greatest reliability (if these do exist), thus further research and optimisation of methodologies and analysis methods are required. With these findings in mind, caution must be exercised before implementing such tools in both amateur and elite sport at this time.

### 4.3. Future Research and Study Limitations

This study was conducted to explore the current status of reliability and diagnostic accuracy of ocular tools for SRCs. It is intended for researchers and practitioners to use this study as a basis to guide future research and development in this area, although this study does have some limitations. Firstly, study quality and risk of bias was not assessed given the exploratory nature of a scoping review, and thus these findings must be interpreted in context of this. A narrow scope of statistical analysis was also utilized in this study. A subsequent systematic review or meta-analysis on this topic may consider additional analysis such as odds ratios, reliable change indexes, and further measures of reliability and diagnostic accuracy. It should also be made clear that the use of a reference standard was used to aid the reader in the interpretation and summarisation of a large dataset across studies; while these are regularly cited, they are very much arbitrary, and some authors have suggested higher thresholds [31,109]. The risk of bias analysis has reflected the subjective nature of much of the literature in this area, where studies are often convenience-sampled and unblinded. This is another potential avenue for emerging technologies to help reduce subjectivity during data collection in such studies.

The authors are also aware of the existence of other emerging ocular technologies being researched both internally by the present research team and externally by others, which were excluded from this study. The rapidly developing nature of technology, machine learning, and artificial intelligence will likely mean that research will continue to accelerate in the coming years; thus, regular revision of the literature may be advised. Limited information is also available on the acceptability of such technologies to date. This is expected but should be considered in subsequent research in this area. Only one included study contained a female-only population, and much of the mixed sex literature was made possible through the well-established US concussion research systems. Future research in less-resourced environments (i.e., amateur female sport), where best practice may not be adhered to [4], and in sports where SRC surveillance may be limited [3], could add additional insights into the utility and acceptability of these tools. At this time, no ocular tool could predict prolonged recovery from SRCs, nor could they be used to determine return to play. Only two studies explored the feasibility of such: ImPACT VMS could not predict duration of recovery (*p* > 0.05), while the KD did not meet RSDA at the commencement of SRC-RTP (*p* > 0.05) [36,102]. This capability is secondary to ensuring accurate initial diagnosis, but if such a tool were developed, it would greatly assist in medical professionals’ confidence in this process.

## 5. Conclusions

SRCs are a challenging but increasingly present injury in amateur and elite sport. Ocular tools may play an important role in assisting with sideline and office-based evaluation of athletes, but each tool may have challenges in achieving standards of reliability and diagnostic accuracy for clinical utility. The KD may be useful in amateur sport, while the VOMs may be useful if reliability is ensured. The NPC assessment should not be used as a standalone assessment for SRCs. For these reasons, the SCAT remains the recommended tool for the assessment of SRCs at this time.

## Figures and Tables

**Figure 1 jfmk-09-00157-f001:**
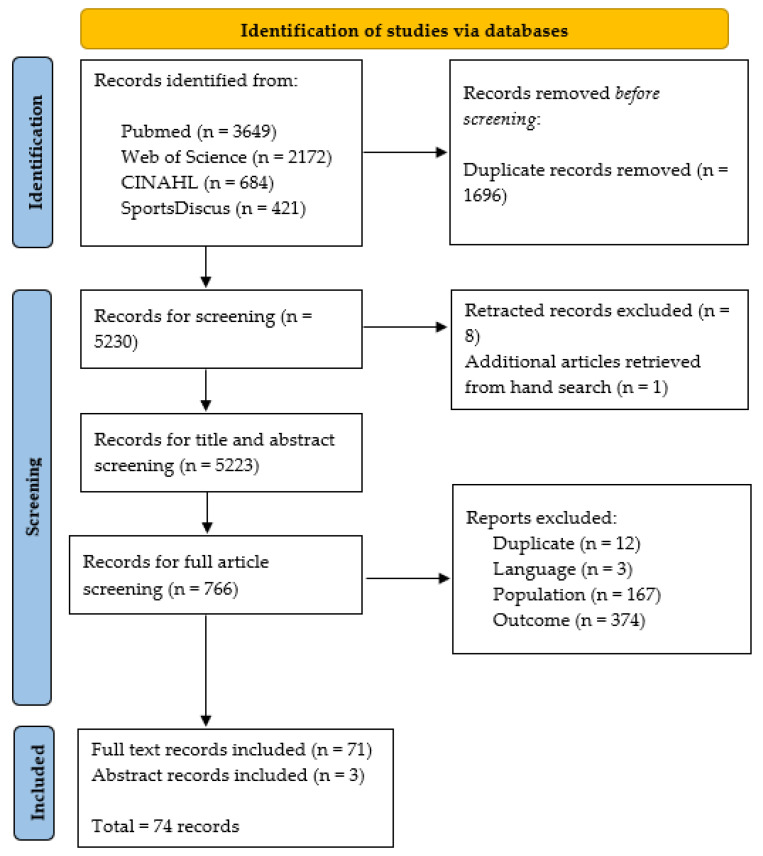
Flow chart of preferred reporting items for systematic review and meta-analysis extension for scoping reviews (PRISMA-SCr).

## Data Availability

All data are available within this manuscript and associated references.

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
