# Peer review of "Clinical Utility of Ocular Assessments in Sport-Related Concussion: A Scoping Review"

_jfmk, 2024, doi:10.3390/jfmk9030157_

Round 1

Reviewer 1 Report (Previous Reviewer 2)

Comments and Suggestions for Authors

To the authors,

               Thank you for undertaking the revisions of this work. The work is much improved and now provides a more consistent and focussed review of this area. This study provides a vital summary of the last decade of research in this area, especially in terms of understanding how useful and reliable (or otherwise) these new technologies are. I can see this being a key reference over the coming years. As such I only have some very, very minor suggestions to make, which should be achievable in the proofing stage.

Well done, and congratulations on a good piece of work!

Kind regards,

INTRODUCTION

Lines 63-66: I like this point you’re trying to make – that these initial studies are the ones that are always cited with few people making use of more up-to-date or more detailed investigations. I think this is an important point to make and one that supports the need for this particular study. Suggest very slight rephrase: “It is common for studies utilising traditional ocular assessments (VOMs and KD) to cite a limited number of seminal studies…….”

MATERIALS & METHODS

Lines 87-88: Suggest making it clear that whilst this previous review investigated variables and measures of interest, it didn’t summarise the reliability of these methods – this again would support the importance of your study.

RESULTS

Having now seen the Supp files I agree with your decision to report the reliability results here rather than in the main file, this makes a lot more sense. In the ‘Reference Standard Met’ column it might be worth considering having a tick for only the studies that met the standard, and leaving the ones that did not meet the standard blank. With such a large table it might make the positive results stand out easier. It might be worth asking a few people not connected with the study to have a look over the table and see whether it would stand out better to them or not.

DISCUSSION

I think the added statements provide a much more consistent narrative, as well as a more focussed exploration of the results and what they mean for us all moving forward in this research area.

Author Response

Dear Reviewer, we appreciate your help in improving this manuscript to date. It has been invaluable and we now believe this manuscript is now ready for publication. Please see attached file for responses.

Reviewer 2 Report (New Reviewer)

Comments and Suggestions for Authors

Detailed review available in the attached report.

Comments on the Quality of English Language

Please revise long sentences for clarity.

Author Response

Dear Reviewer, thank you for your support it greatly improving this manuscript. Your insight have been invaluable, we believe this updated manuscript is now ready for publication (pending proofing and journal re-formatting). Please see attached file for responses!

This manuscript is a resubmission of an earlier submission. The following is a list of the peer review reports and author responses from that submission.

Round 1

Reviewer 1 Report

Comments and Suggestions for Authors

This is a review article aiming to summarize the available ocular-based field tests/instrument and evaluate their clinimetric properties to diagnose sport-related concussions. Some diagnostic tools appear to use a dichotomous outcome (presence/absence or positive/negative), while others a continuous outcome or even some raters’ agreement (qualitative assessment?). Authors assessed the diagnostic performance of the former by means of diagnostic accuracy metrics whereas for continuous scoring some other corresponding metrics (e.g., reliability statistics). Authors have not clearly differentiated between those type of outcomes leaving readers with less interpretability skills to appraise the instruments and the authors reasoning/arguments.

The study objectives of this review are within the territory of systematic reviews (1), and I can’t really understand why a scoping review approach was selected instead. PRISMA has guidelines for conducting reviews on diagnostic accuracy studies (PRISMA–DTA) and on reliability or measurement error of outcome measurement instruments. Risk of bias evaluation is an important part of appraising the quality of the studies that evaluated diagnostic performance of tests/tools/instruments and hence the clinimetric importance of those tools, but authors have not used the QUADAS 2 or COSMIN risk of bias assessment to make judgements. I disagree on the arguments presented in lines 131–133. Providing an overview of the diagnostic tools and their performance without any judgment on the quality of the studies they are grounded is not a good argument for conducting a scoping review instead. The results, discussion and conclusions of their study may be highly biased, and therefore of questionable trustworthiness.

As an additional remark, authors frequently refer reviewers/readers to supplementary material and appendixes, but, except for Appendix 1, none are  available for appraisal.

1– Munn, Z., Peters, M. D. J., Stern, C., Tufanaru, C., McArthur, A., & Aromataris, E. (2018). Systematic review or scoping review? Guidance for authors when choosing between a systematic or scoping review approach. BMC medical research methodology18(1), 143. https://doi.org/10.1186/s12874-018-0611-x

Author Response

Comment: This is a review article aiming to summarize the available ocular-based field tests/instrument and evaluate their clinimetric properties to diagnose sport-related concussions. Some diagnostic tools appear to use a dichotomous outcome (presence/absence or positive/negative), while others a continuous outcome or even some raters’ agreement (qualitative assessment?). Authors assessed the diagnostic performance of the former by means of diagnostic accuracy metrics whereas for continuous scoring some other corresponding metrics (e.g., reliability statistics). Authors have not clearly differentiated between those type of outcomes leaving readers with less interpretability skills to appraise the instruments and the authors reasoning/arguments.

Response: Thank you for taking the time to review our study, we really do appreciate your input. Regarding the statistics utilized in this study, we have now clarified each statistic in line 140 ‘Charted data was supported…’. To be clear, we have not chosen reliability statistics for some tools and diagnostic statistics for other tools – we ideally wanted to present all data on the reliability and diagnostic accuracy of all of these tools. In some cases this was possible, in others it was not.

Comment: The study objectives of this review are within the territory of systematic reviews (1), and I can’t really understand why a scoping review approach was selected instead. PRISMA has guidelines for conducting reviews on diagnostic accuracy studies (PRISMA–DTA) and on reliability or measurement error of outcome measurement instruments. Risk of bias evaluation is an important part of appraising the quality of the studies that evaluated diagnostic performance of tests/tools/instruments and hence the clinimetric importance of those tools, but authors have not used the QUADAS 2 or COSMIN risk of bias assessment to make judgements. I disagree on the arguments presented in lines 131–133. Providing an overview of the diagnostic tools and their performance without any judgment on the quality of the studies they are grounded is not a good argument for conducting a scoping review instead. The results, discussion and conclusions of their study may be highly biased, and therefore of questionable trustworthiness.

Response: I think we can absolutely understand your point of view on both points here. Our scoping review was iterative and intends to provide a broad overview of the current literature and assess gaps in our current knowledge. Therefore we proceeded with a scoping review, what we felt was most appropriate for the study and our needs.

Regarding the risk of bias, we agree risk of bias is relevant to this study and have since included both the COSMIN and QUADAS-2 tools (See Table 1). As you will see within the study, it is difficult to use both tools in this area given the subjectivity of the assessments/research in this area, which we have now flagged as another relevant finding. Thank you for your support in improving the study in this aspect.

Comment: As an additional remark, authors frequently refer reviewers/readers to supplementary material and appendixes, but, except for Appendix 1, none are  available for appraisal.

Response: We apologize for the absence of these within the submission, I am unsure what happened but they are included within the draft now. Again, our sincerest apologies for this.

Reviewer 2 Report

Comments and Suggestions for Authors

To the authors,

               Thank you for completing this work and submitting it for review. This is a vital topic at the moment, and will almost certainly remain so for many years, so having this kind of summary available for researchers and practitioners would be of great benefit. There are currently some issues with the presentation of the data, the consistency of the presented aims, and the lack of focus or depth in the discussion. I think there is quite a bit of work still to do before this is ready for publication, but I very much look forward to reviewing the resubmitted version of this work as I think it would end up being an important piece. Please take the following comments with the support they are intended.

Kind regards,

Please check for spelling and grammar errors throughout.

INTRODUCTION

Line 34: This is a small point, but I suggest rephrasing to avoid the term ‘gold standard’, as the SCAT is only capable of - and should only be used as a method for - determining that brain dysfunction has occurred and that further medical attention is required. An MRI or CT scan would be the ‘gold standard’ as these are methods of directly measuring brain injury. Suggest stating something along the lines that the SCAT is the most commonly applied method in a sporting setting/SRC research.

Line 50-52: Unsure what this sentence means – please rephrase to make the meaning clearer.

Lines 53-61: This is a good paragraph to set up the aims and contents of the manuscript – but have the reliability/validity of the tools discussed been questioned previously? If so, then please provide this information here to better support the aims of this study and set up the narrative better.

METHODS

Lines 74-75: This statement differs quite a lot to the stated aim of the study in lines 62-64. Summarising the reliability and accuracy of these tools is not the same as supporting their use for diagnosis of SRCs or return to play. Please reword either/both of these statements to more accurately represent what has been attempted in this study.

Lines 80-81: If a systematic review already exists on this topic, what is the added value of this study? Please provide a justification of what this manuscript adds to the previous review in the Introduction section.

Lines 127-128: Appendix 2 has not been included in the manuscript.

Line 134: Does ‘charted’ mean ‘reported’ here?

Lines 150-151: Have p values been calculated for any of the data by the authors of this manuscript? If not, please remove.

RESULTS

Tables 1-3: Please include a column/s reporting the key statistic/s for each study (i.e., the ICC, the AUC, etc.). I realise these might be reported in the supplemental files, but I don’t have access to these to check, and given that these are the key data on which the manuscript relies, it would be better to have them readily available in the tables of the main manuscript – having these data easily accessible in one place is where the real value of this paper will be to the reader.

Lines 155-156: The reported numbers in these lines differ to those shown in Figure 1. Please check and correct whichever ones are incorrect.

Lines 162-163; 171-172: This may be a handling editor error, but Supplement 1, Supplement 2.1 and Supplement 2.2 have not been provided for review.

Line 268: Please state exactly how many studies used ‘emerging technologies’ – it might be worthwhile defining what an ‘emerging technology’ is in the context of this work in the Introduction section.

DISCUSSION

Lines 294-296: Please refer to previous comments on Lines 74-75 regarding the consistency and specificity of the stated aims.

Lines 300-329: This currently reads like a shorter summary of the results rather than a statement of what the key findings were. Please re-write to provide a more useful summary for the reader. What were the key ‘big picture’ outcomes of this work? Are these tools useful or not?

Lines 327-329: I’m unsure what this sentence means – what are the ‘alternative tools and technologies’? What does this refer to in relation to this study? Which specific supplementary file? In what way were they included or exploratory?

Lines 342-344: what is meant by ‘rudimentary’ here? Why would this be a problem in elite/semi-pro sport and not others?

Line 356: External pressures related to what? Please ensure the thoughts and meanings of each statement have been fully realised – don’t leave it to the reader to guess the meaning or intention of the statements.

Lines 363-367: This is the key findings and discussion point of the study, and requires much more focussed attention in this discussion section. This needs expanding on, including discussion of why this might be the case – what are these methods actually measuring? Are they direct, indirect or proxy measures of brain injury? Are they actually measuring anything to do with the brain at all? What is the problem with measuring these things in the real world? Why are these measurements so varied and ‘unreliable’? These points should be the focus of this entire discussion.

Lines 367-369: I’m unsure of the meaning of this sentence – is this further research relevant? If it is then why not discuss it in detail here?

Lines 384-386: Unsure what is meant here by ‘internally and externally’, please rephrase.

Lines 393-395: This is another key finding that hasn’t really been discussed or expanded on in any detail. I’d suggest making more of this key finding.

CONCLUSIONS

Given the uncertain reliability and validity of these methods that you’ve presented here, I’d suggest this needs to be a more explicit point in this section. Please see previous comments about whether the SCAT is a ‘gold standard’ and whether this phrase might need to be changed here.

Comments on the Quality of English Language

Please check throughout for minor spelling and grammatical errors.

Author Response

Response to Reviewer 2

Comment: To the authors, Thank you for completing this work and submitting it for review. This is a vital topic at the moment, and will almost certainly remain so for many years, so having this kind of summary available for researchers and practitioners would be of great benefit. There are currently some issues with the presentation of the data, the consistency of the presented aims, and the lack of focus or depth in the discussion. I think there is quite a bit of work still to do before this is ready for publication, but I very much look forward to reviewing the resubmitted version of this work as I think it would end up being an important piece. Please take the following comments with the support they are intended. Kind regards.

Response: Thank you for taking the time to review our manuscript, your feedback has been insightful and we hope we have addressed many of your comments sufficiently below.

INTRODUCTION

Comment: Line 34: This is a small point, but I suggest rephrasing to avoid the term ‘gold standard’, as the SCAT is only capable of - and should only be used as a method for - determining that brain dysfunction has occurred and that further medical attention is required. An MRI or CT scan would be the ‘gold standard’ as these are methods of directly measuring brain injury. Suggest stating something along the lines that the SCAT is the most commonly applied method in a sporting setting/SRC research.

Response: Agreed. This has been amended to read: ‘The recommended SRC assessment of athletes at this time is the Concussion in Sport Group’s Sport Concussion Assessment Tool 6 (SCAT6)’.

Comment: Line 50-52: Unsure what this sentence means – please rephrase to make the meaning clearer.

Response: Agreed. Clarified as: ‘The associated eye movements (e.g. saccades and smooth pursuits) of these tests are a key focus of emerging ocular technologies in SRC research. [17,18]’.

Comment: Lines 53-61: This is a good paragraph to set up the aims and contents of the manuscript – but have the reliability/validity of the tools discussed been questioned previously? If so, then please provide this information here to better support the aims of this study and set up the narrative better.

Response: Agreed. One of the key issues with literature in this area is many studies cite the 1-2 seminal studies with these tools for their reliability or accuracy statistics, hence the usefulness of this study to provide a broader summary of the available data. As such the following has been included: More often than not, studies utilizing traditional ocular assessments (VOMs and KD) cite 1-2 seminal studies for reliability or diagnostic accuracy statistics,[14,15,22] while little is known about the reliability and diagnostic accuracy of emerging ocular technologies.’

METHODS

Comment: Lines 74-75: This statement differs quite a lot to the stated aim of the study in lines 62-64. Summarising the reliability and accuracy of these tools is not the same as supporting their use for diagnosis of SRCs or return to play. Please reword either/both of these statements to more accurately represent what has been attempted in this study.

Response: Agreed. Reworded to: ‘The key research question supporting this study was: Is there research to support the use of ocular assessment tools commonly used to diagnose SRCs in athletes? More specifically this study had three questions to answer: Which ocular tools provide (1) optimal reliability, (2) internal consistency, and (3) diagnostic accuracy in the assessment and diagnosis of SRCs.’

Comment: Lines 80-81: If a systematic review already exists on this topic, what is the added value of this study? Please provide a justification of what this manuscript adds to the previous review in the Introduction section.

Response: Agreed. This has been amended in Methods to: ‘Initial search terms and strings were conducted by AW and were based on a previous systematic review on ocular technologies for SRCs which investigated variables and measures of interest for future research.[25]’

Comment: Lines 127-128: Appendix 2 has not been included in the manuscript.

Response: We apologize for any errors in this regard. This has now been provided.

Comment: Line 134: Does ‘charted’ mean ‘reported’ here?

Response: Charted is the preferred term for us here.

Comment: Lines 150-151: Have p values been calculated for any of the data by the authors of this manuscript? If not, please remove.

Response: Apologies. This is relevant for appendixed data.

RESULTS

Comment: Tables 1-3: Please include a column/s reporting the key statistic/s for each study (i.e., the ICC, the AUC, etc.). I realise these might be reported in the supplemental files, but I don’t have access to these to check, and given that these are the key data on which the manuscript relies, it would be better to have them readily available in the tables of the main manuscript – having these data easily accessible in one place is where the real value of this paper will be to the reader.

Response: The appendixed files have now been provided. I hope you can see the large amount of data which is available and thus the supplementary files were deemed necessary.

Comment: Lines 155-156: The reported numbers in these lines differ to those shown in Figure 1. Please check and correct whichever ones are incorrect.

Response: These are correct. 5223 articles were screened, 71 (articles) + 3 (abstracts) equal 74 studies included in the review.

Comment: Lines 162-163; 171-172: This may be a handling editor error, but Supplement 1, Supplement 2.1 and Supplement 2.2 have not been provided for review.

Response: As above these are now provided, we apologize for any inconvenience.

Comment: Line 268: Please state exactly how many studies used ‘emerging technologies’ – it might be worthwhile defining what an ‘emerging technology’ is in the context of this work in the Introduction section.

Response: Restated as: ‘Five studies used emerging technologies to assess saccades and smooth pursuits (not including KD eye-tracking previously stated)’. Definition now included within the introduction: ‘Emerging technologies included new, untested technologies or technologies still within their infancy in research or practice.’

DISCUSSION

Comment: Lines 294-296: Please refer to previous comments on Lines 74-75 regarding the consistency and specificity of the stated aims.

Response: Clarified each within the intro/method/discussion aims.

Comment: Lines 300-329: This currently reads like a shorter summary of the results rather than a statement of what the key findings were. Please re-write to provide a more useful summary for the reader. What were the key ‘big picture’ outcomes of this work? Are these tools useful or not?

Response: This paragraph was intended to simply summarize the study given the extent of data available, I have now added an initial clarification of key findings and re-iteration of such at the end of each paragraph on KD, VOMs/NPC, and alternative tools and technologies.

Comment: Lines 327-329: I’m unsure what this sentence means – what are the ‘alternative tools and technologies’? What does this refer to in relation to this study? Which specific supplementary file? In what way were they included or exploratory?

Response: I’ve reworded this sentence as follows: ‘Additional information regarding the remaining alternative tools and technologies (as per Table 3) is available in the supplementary table 2.3. However, many of these studies were exploratory or validation studies and none had both of the required RSR and RSDA.’

Comment: Lines 342-344: what is meant by ‘rudimentary’ here? Why would this be a problem in elite/semi-pro sport and not others?

Response: Clarified the statement and added a reference for the poor attitudes statement: ‘This may perhaps be linked to the rudimentary scoring system employed with the tool (total time) enabling inflated baselines from athletes in elite male athletes where poor attitudes towards this injury are prevalent.[45,106]’

Comment: Line 356: External pressures related to what? Please ensure the thoughts and meanings of each statement have been fully realised – don’t leave it to the reader to guess the meaning or intention of the statements.

Response: Clarified statement as: ‘Achieving truly objective assessment is a pertinent point as athletes may attempt to falsify assessment,[107] while allied medical staff may feel pressure and bias from coaches or athletes during SRC assessments.[4]’.

Comment: Lines 363-367: This is the key findings and discussion point of the study, and requires much more focussed attention in this discussion section. This needs expanding on, including discussion of why this might be the case – what are these methods actually measuring? Are they direct, indirect or proxy measures of brain injury? Are they actually measuring anything to do with the brain at all? What is the problem with measuring these things in the real world? Why are these measurements so varied and ‘unreliable’? These points should be the focus of this entire discussion.

Response: I’ve tried to clarify this statement, at this time a lot of the inaccuracies in the eye tracking technologies is because they are still under development as evidenced by the data provided in supplementary table 2.3, but I have also added additional reasoning: ‘Why eye-tracking technologies have not been successful to date is unclear. It should be stated firstly that all oculomotor tools are indirect measures of brain function and thus are not as accurate as a more direct measure of brain injury.  It could also be suggested that not all athletes who sustain an SRC present with oculomotor deficits, therefore impacting the diagnostic accuracy of such devices. As evidenced by the findings of these studies in Supplementary table 2.3, it is as of yet unclear which variables provide the greatest reliability (if these do exist), thus further research and optimisation of methodologies and analysis methods are required. With these findings in mind, caution must be exercised before implementing such tools in both amateur and elite sport at this time’.

Comment: Lines 367-369: I’m unsure of the meaning of this sentence – is this further research relevant? If it is then why not discuss it in detail here?

Response: Clarified the statements relevance: ‘The review did however suggest that challenging measures of executive function (i.e. memory-based saccades, antisaccades) may be a promising area for future research. Subsequent research by the same author has been included in the present study and is available in Supplement 2.3. The study found antisaccades using an emerging technology achieved moderate to good reliability in youth and adult participants, but many other explored variables performed poorly.[93] With these findings in mind, caution must be exercised before implementing such tools in both amateur and elite sport at this time.’

Comment: Lines 384-386: Unsure what is meant here by ‘internally and externally’, please rephrase.

Response: Clarified as: ‘The authors are also aware of the existence of other emerging ocular technologies being researched both internally by the present research team and externally by others which were excluded from this study.’

Comment: Lines 393-395: This is another key finding that hasn’t really been discussed or expanded on in any detail. I’d suggest making more of this key finding.

Response: There was very little data available but statement has been clarified: ‘At this time, no ocular tool could predict prolonged recovery from SRCs nor could they be used to determine return to play. Only two studies had explored the feasibility of such, Impact VMS could not predict duration of recovery (p > 0.05) while the KD did not meet RSDA at the commencement of SRC-RTP (p > 0.05).[36,102] This capability is secondary to ensuring accurate initial diagnosis, but if such a tool was developed it would greatly as-sist in medical professionals confidence in this process.’

CONCLUSIONS

Comment: Given the uncertain reliability and validity of these methods that you’ve presented here, I’d suggest this needs to be a more explicit point in this section. Please see previous comments about whether the SCAT is a ‘gold standard’ and whether this phrase might need to be changed here.

Response: Paragraph has been rewritten as: ‘SRCs are a challenging but increasingly present injury in amateur and elite sport. Ocular tools may play an important role in assisting in sideline and office-based evaluation of athletes but each tool may have challenges in achieving standards of reliability and diag-nostic accuracy for clinical utility. The KD may be useful in amateur sport, while the VOMs may be useful if reliability is ensured, the NPC assessment should not be used as a standalone assessment for SRCs. For these reasons, the SCAT remains the recommended tool for the assessment of SRCs at this time.’

Comment: Please check throughout for minor spelling and grammatical errors.

Response: This has been completed. Thank you.